**Data Availability Statement:** Data cannot be shared publicly because access to individual-level

# Labour market marginalisation in young refugees and their majority peers in Denmark and Sweden: The role of common mental disorders and secondary school completion

Christopher Jamil de Montgomery[1,2]*, Marie Norredam[1,3], Allan Krasnik[1], Jørgen Holm Petersen[4], Emma Björkenstam[2,5], Lisa Berg[6], Anders Hjern[6], Marit Sijbrandij[7], Peter Klimek[8,9], Ellenor Mittendorfer-Rutz[2,8]

1 Department of Public Health, Danish Research Centre for Migration, Ethnicity and Health (MESU), University of Copenhagen, Copenhagen K, Denmark, 2 Department of Clinical Neuroscience, Division of Insurance Medicine, Karolinska Institutet, Stockholm, Sweden, 3 Section of Immigrant Medicine, Department of Infectious Diseases, University Hospital Hvidovre, Copenhagen, Denmark, 4 Department of Biostatistics, University of Copenhagen, Copenhagen, Denmark, 5 Department of Neuroscience, Psychiatry, Uppsala University, Uppsala, Sweden, 6 Center for Health Equity Studies, Stockholm University, Stockholm, Sweden, 7 Department of Clinical, Neuro and Developmental Psychology, World Health Organization Collaborating Center for Research and Dissemination of Psychological Interventions, Amsterdam Public Health research institute, Vrije Universiteit Amsterdam, The Netherlands, 8 Section for Science of Complex Systems, CeMSIIS, Medical University of Vienna, Vienna, Austria, 9 Complexity Science Hub Vienna, Vienna, Austria

* cmon@sund.ku.dk

## Abstract

### Background

Due to the circumstances of their early lives, young refugees are at risk of experiencing adverse labour market and health outcomes. The post-settlement environment is thought to play a decisive role in determining how this vulnerability plays out. This study compared trends in labour market marginalisation in young refugees and their majority peers during early adulthood in two national contexts, Denmark and Sweden, and explored the mediating role of common mental disorders and secondary school completions.

### Methods

Using registry data, 13,390/45,687 refugees were included in Denmark/Sweden and 1:5 matched to majority peers. Inequalities in labour market marginalisation were investigated during 2012–2015 in each country using linear probability models and mediation analysis. Country trends were standardised to account for differences in observed population characteristics.

### Results

The risk of marginalisation was 2.1–2.3 times higher among young refugees compared with their majority peers, but the risk decreased with age in Sweden and increased in Denmark for refugees. Birth-cohort differences drove the increase in Denmark, while trends were

administrative data in Denmark and Sweden is restricted by national legislation to specific projects at approved research institutions and is subject to an application process directly to the responsible agencies. For further information, please refer to the webpages of Statistics Denmark (www.dst.dk) and Statistics Sweden (www.scb.se).

**Funding:** This study was funded by a grant from the Swedish Research Council (https://www.vr.se/english.html), Grant number: 2018-05783. The grant was awarded to senior co-author Ellenor Mittendorfer-Rutz (EM). The funders had no role in study design, data collection and analysis, decision to publish, or preparation of the manuscript.

**Competing interests:** The authors have declared that no competing interests exist.

consistent across birth-cohorts in Sweden. Differences in population characteristics did not contribute to country differences. Common mental disorders did not mediate the inequality in either country, but secondary school completions did (77–85% of associations eliminated).

## Conclusions

The findings document both the vulnerability of young refugees to labour market marginalisation and the variability in this vulnerability across post-settlement contexts. While the contrast in policy climates in Denmark and Sweden sharpened over time, the risk of marginalisation appeared more similar in younger cohorts, pointing to the importance of factors other than national immigration and integration policies. Institutional efforts to assist young refugees through secondary education are likely to have long-lasting consequences for their socio-economic trajectories.

## Introduction

The consequences of mental ill-health during the transition to adulthood can be long-lasting, as inequalities that crystallise during the transition to adulthood tend to foreshadow socio-economic and health disparities that persist and worsen throughout life [1, 2]. Refugee children stand out as a population of special concern in this regard. Across refugee groups and host societies, refugee children have generally been found to have elevated levels of mental ill-health, including anxiety, depression and stress-related disorders, as well as high levels of externalising and internalising behaviour [3, 4]. As newcomers, they frequently have to acquire one or several new languages while navigating unfamiliar educational settings, but adverse circumstances during their early lives also frequently expose them to stressful and traumatising experiences and protracted periods without access to schooling.

At the same time, a number of longitudinal studies have shown that, as a population, refugee children have great capacity for resilience, i.e. "a good mental health outcome following an adverse life event or a period of difficult life circumstances" [5], and that their mental health tends to improve notably following resettlement [6–8]. In fact, stable settlement conditions and positive school experiences have been identified as crucial in this regard [9–11], while the aggravating stress of prolonged periods of uncertainty and discrimination undermines this capacity [12–16]. So while inequalities between refugees and their majority peers are indicative of differences in preconditions, they are simultaneously indicative of the capacity of host-societies to provide refugee children with the necessary opportunities and resources. In this way, integration processes can be situated within a broader agenda of social inequality by focusing on the critical question of how refugee-hosting societies can provide forcibly displaced children and youth with the best possible opportunities in life.

This sets the stage for comparative research. Comparative integration context theory argues that institutional arrangements pertaining to educational systems, labour markets, health services, and other domains of social life, correlate with different life course patterns in immigrant populations across Europe [17]. Institutional factors such as ability tracking, alternative pathways to higher education, second-language support at different educational levels, and schooling options before and after compulsory school age all shape the educational pathways of refugee children [18], even as restrictive immigration policies have been argued to exacerbate

health vulnerabilities in migrant populations more generally [19, 20]. The concept of 'refugee-competent schools' [21, 22], i.e. schools capable of fostering psycho-social well-being among young refugees as well as educational attainment, has been coined in the qualitative literature on young refugees' school experiences to unpack the features of schools that provide those positive experiences for young refugees that predict future health and socio-economic outcomes. Attention towards capacity development at schools to foster such competencies may also vary between countries, as priorities at various levels of administration differ. While being an important predictor of future labour market exclusion in its own right [23], independent of mental health, educational attainment is then also a potential mediator on the pathway between mental health challenges in young refugees and future labour market outcomes.

The comparison of Denmark and Sweden is an illustrative case. As Nordic welfare states, both countries are characterised by comprehensive public services financed by broad taxation, including general health services and free of charge education from the primary to tertiary level, but within the domain of immigration and integration policy they diverge in the paths taken since the 1980s [24]. The 'restrictive' trend in Danish immigration policy is expressed in restrictions to immigrants' access to financial support, family reunification, and possibilities for naturalization, trailing well behind Sweden at the top of the Migration Integration Policy Index [25]. It is also expressed in the frequency of changes made to the Danish immigration legislation. On average, the Danish Immigration Act was amended every three months between 2002 and 2016 [26]. Among these changes, the rights concerning permanent residence of individuals who have been granted asylum in Denmark were changed ten times between 2007 and 2017 [27]. In discursive terms, the framing of the issue of migration from state institutions in Sweden tends to emphasise cultural self-determination and equal opportunities, in contrast to a framing in Denmark centred on the challenges to social order presented by diverging cultures and migrants' unwillingness to adapt to a Danish model of life [28]. In terms of scale, immigration to Sweden started earlier and has been larger in size and by the 2010s, one in five Swedish residents was either born abroad or a child of parents born abroad, compared with one in ten in Denmark [29]. Of the half million refugees who resettled in the Nordic countries during 2000–2016, 61% settled in Sweden and 13% settled in Denmark [30].

Few studies have explored the educational and labour market outcomes in young refugee populations in the Nordic countries [30]. A comparative report from 2020 found that refugee children resettled 1986–2005 fared better across a range of educational, employment and health outcomes in Sweden than in Denmark [31]. However, the consequences of differences in the composition of each country's refugee population for country-level differences in outcomes were not addressed. In addition, the comparative role of mental ill-health in social inequalities has not been explored, although studies within particular Nordic countries have shown a strong relationship between common mental disorders and labour market marginalisation, including long-term unemployment and disability pension, in both immigrant and native populations of youth [32]. Finally, the development of social inequalities over biographical time is not typically addressed and disentangled from changes over calendar time. Without such disentanglement, it is not possible to gauge whether the life chances of young refugees within a given context change over time; a question with important policy implications that is analytically distinct from the question of how inequalities unfold over the life span.

This study adds to current knowledge by addressing these gaps of evidence. Considering the risk of labour market marginalisation among refugee youth and their majority peers living in Denmark and Sweden during 2012–2015, the study sheds light on 1) how social inequality developed over four years during the transition to adulthood for different birth cohorts and 2) the mediating role of common mental disorders and secondary school completion in this inequality.

## Materials and methods

The analysis utilised national registry data from Denmark and Sweden linked at the individual level.

### Study population

All residents in Denmark/Sweden on 1 January/31 December each year during 2010-2012/ 2009-2011, who were aged 19–25 in 2011 (born 1986–1993), were included if they (a) immigrated to the country in or before 2009 as refugees (obtaining refugee status themselves or through family reunification to a refugee) and (b) were below the age of 18 at settlement (refugee population; $N_{DK}$ = 13,390/ $N_{SE}$ = 45,687). In both countries, refugee status was indicated through information on grounds of residence collected by the national immigration agencies (in Denmark in combination with country-of-origin information). A 1:5 matched sample of individuals born in Denmark/Sweden with at least one parent also born in Denmark/Sweden was included (majority population; $N_{DK}$ = 69,650/ $N_{SE}$ = 227,287), matched on year of birth, sex and the type of domicile municipality in 2010.

### Labour market marginalisation

Dichotomous indicators for labour market marginalisation (LMM) each year from 2012–2015 were coded with information about labour market income, receipt of government stipends, registered unemployment, as well as income from disability pensions, providing up to four observations per individual. Labour market income included salaries, surplus income from self-employment, the value of employee perks, as well as maternity and paternity leave benefits and sickness benefits paid by the employer.

We followed the approach of Bäckman et al. [23], operationalising LMM as non-students with labour market inactivity indicated by a labour market income of less than 12.5% of the median labour market income among the population aged 20–64 years. In Bäckman et al.'s definition, 12.5% of the median labour market income is a pragmatic cut-off between labour market positions defined as no or negligible labour market participation and part-time or unstable work. While a full-time income at a low salary corresponds approximately to 87.5% of the median labour market income, 12.5% represents an annual income below what would be earned in 2 months in such a position. Additionally, those with an income from disability pensions of at least 25% of the median labour market income (which corresponds to approximately half a year of full-time disability pension) and those registered as unemployed for at least 180 days during the year (more than half the year) were also included in the LMM category. Although the precise cut-off values to a certain extent are arbitrary, these definitions have been used in several other Nordic comparative studies of labour market marginalization [31, 33, 34] and aligning the definitions across studies facilitates comparisons. Recipients of stipends amounting to at least 8% of the median labour market income were classified as students. This threshold resulted in a similar proportion of students as in official enrolment statistics. However, youth still in secondary education appeared to be less effectively identified than those in higher education, especially in Sweden. In Denmark, where enrolment data was available, we conducted a sensitivity analysis using this data.

### Main variables

Common mental disorders (CMD) during 2009–2011 were defined as either inpatient or specialised outpatient care with a main diagnosis related to depressive disorders (ICD-10 codes F32-33), phobic anxiety disorders (F40), other anxiety disorders (F41), obsessive-compulsive

disorder (F42), or disorders related to reactions to severe stress (F43); or purchase of a prescribed antidepressants (Anatomical Therapeutic Chemical classification, code N06A). As a sensitivity analysis, we also investigated whether contact patterns differed between the populations when considering hospital contacts (in- or outpatient) and antidepressant purchases separately.

Secondary school completion was measured annually and was defined as a registered highest level of education corresponding to the International Standard Classification of Education level 3 or higher.

## Other covariates

Demographic information concerning year of birth, sex and municipality type in 2010 (coded as either city (the capital area in Denmark, in Sweden capital area plus Gothenburg and Malmo), large town (at least 90,000 inhabitants), or smaller town (less than 90,000 inhabitants)) was included for all individuals. For refugees, we further included information on age at immigration, i.e. age when granted refugee status (in categories: preschool (0–5 years old), early school (6–10 years old), late school (11–15 years old), and post-compulsory school (16–17 years old)); country/region of origin (in categories: former Yugoslavia, Iran, Iraq, Afghanistan, Somalia, and "others"); and accompanying family at arrival (in categories: both parents, one parent, no parents) which was coded based on information on date of arrival and family ties in both countries.

## Analytical procedures

Analyses were conducted separately in each country. First, the proportion of individuals in each population group who experienced LMM each year from age 20 to age 29 was calculated and presented graphically. To address the possibility of compositional differences (especially among refugees) in Denmark and Sweden, we also plotted standardised results. The standardisation proceeded as follows: first linear probability models were estimated in Sweden (separately for refugees and the majority population) from age 20–29. The estimated coefficients ($\hat{B}_{SE}$) were then exported to the Danish dataset and multiplied with the Danish covariates at their means ($\bar{X}_{DK}$) to produce predicted probabilities. These can be interpreted as the proportion of youth who would have experienced LMM given the Swedish probabilities and the Danish distribution across covariates, with the simplifying assumption that interaction effects did not differ importantly between the countries. Standard errors were estimated as $[\bar{X}_{DK} \, V(\hat{B}_{SE}) \bar{X}_{DK}]$, where $V(\hat{B}_{SE})$ refers to the cluster robust variance-covariance matrix for the parameters estimated in Sweden, accounting for repeated observations of the same individuals. Confidence intervals were included for the standardised lines, but not for the raw proportions. As the included confidence intervals illustrate, the large sample sizes led to very narrow confidence intervals, and for the refugee samples proportions were calculated on the full populations.

Second, the proportion of the relative difference between refugee and majority youth in LMM which was mediated by a) common mental disorders and b) secondary school completion was estimated following the approach to mediation analysis outlined in [35]. The total effect was defined as the odds-ratio of LMM of refugees compared with their majority peers and was estimated using logistic regression with clustered standard errors to account for repeated observations. The conditional direct effect was estimated similarly but weighted by the inverse probability of the mediator weights (IPMW). The IPMW were calculated by first fitting a logistic regression of the mediator on population group, birth year, municipality type and sex, and then dividing the overall probability of the mediator by the predicted probabilities

for each individual. Finally, the portion eliminated was estimated offsetting by the controlled direct effect, while the proportion eliminated was calculated as [(total effect–controlled direct effect)/(total effect-1)].

### Ethics approval

All data were fully anonymized by Statistics Denmark and Statistics Sweden before we accessed them. Processing of the Swedish data was approved by the regional ethical committee in Stockholm (dnr: 2007/762-31). In Denmark, ethical committee approval for studies using administrative data is not required, but data processing was authorised by the Office of Research and Innovation at the Faculty of Health and Medical Sciences at the University of Copenhagen (case number SUND-2016-65). In accordance with Danish and Swedish legislation concerning research using registry data, informed consent was not required for this study.

## Results

### Inequalities in labour market marginalisation

Tables 1 and 2 show the distribution across the included variables among refugees and the majority population in Denmark and Sweden. In both Denmark and Sweden, the proportion of youth with CMD was larger in the majority population than in the refugee population. This pattern was consistent whether CMD was defined based on hospital contacts alone, purchases of antidepressants alone, or both combined (not shown). The difference was larger among girls than boys in both countries. Levels of completed secondary school were higher in Sweden than Denmark, but levels increased in both countries over time in all population groups. In both Denmark and Sweden, the level of completions was around 16 percentage points lower among refugee youth than majority youth in 2012. By 2015, the difference diminished to around 11 percentage points in Sweden while it increased to around 19 percentage points in Denmark.

Fig 1 shows the proportion in each population group who experienced LMM from age 20–29 in Denmark and Sweden, as well as the standardised proportions. In both Denmark and Sweden, the levels of LMM were higher among refugees than among the majority population. However, in Denmark LMM increased for refugees from around age 23 (from 21.7 pct. at age 23 to 26.8 pct. at age 29), while it decreased throughout the age-range for refugees in Sweden (reaching 18.6 pct. at age 29). The difference between the Swedish proportions and the standardised proportions were negligible and given the sample size the confidence intervals for the standardised lines were very narrow. We expected that misclassification due to secondary school enrolments in Sweden would bias the proportions upward at younger ages, and the dramatic decrease at ages 20–22 in Sweden point to this being the case. With this in mind, Fig 1 suggests that the differences between refugees in Denmark and Sweden were small during ages 20–25 and then increased (from similar or a few percentage points lower proportions in Denmark during the early 20s to 8.3 percentage points higher proportions at age 29).

In Figs 2 and 3, the proportions were disaggregated by CMD and completed upper secondary education, respectively. Each birth cohort was plotted with separate lines to tease apart the development over age from the development between birth cohorts. In both countries, the level of LMM was higher among those with CMD than among those without, and lower among those with education than among those without; and this was so for both refugees and the majority population. While levels of LMM appeared to increase in Denmark from age 23, Figs 2 and 3 show that it was driven by differences between birth cohorts. The figures also shows that in Sweden, the downward trend in LMM over age was present among both refugees

**Table 1. Descriptive statistics of time invariant covariates among young refugees aged 19–25 in 2012 and their majority peers.**

| | DENMARK | | | | SWEDEN | | | |
|---|---|---|---|---|---|---|---|---|
| | Refugees | | Majority | | Refugees | | Majority | |
| | No. | % | No. | % | No. | % | No. | % |
| **Total** | 13,930 | 100 | 69,650 | 100 | 45,687 | 100 | 227,287 | 100 |
| **Common mental disorder** | | | | | | | | |
| Yes | 963 | 6.9 | 6,555 | 9.4 | 2,854 | 6.2 | 19,413 | 8.5 |
| No | 12,967 | 93.1 | 63,095 | 90.6 | 42,833 | 93.8 | 207,874 | 91.5 |
| **Sex** | | | | | | | | |
| Male | 7,704 | 55.3 | 38,520 | 55.3 | 24,708 | 54.1 | 122,962 | 54.1 |
| Female | 6,226 | 44.7 | 31,130 | 44.7 | 20,979 | 45.9 | 104,325 | 45.9 |
| **Birth cohort** | | | | | | | | |
| 1986 | 1,821 | 13.1 | 9,105 | 13.1 | 5,972 | 13.1 | 29,664 | 13.1 |
| 1987 | 1,881 | 13.5 | 9,405 | 13.5 | 5,974 | 13.1 | 29,689 | 13.1 |
| 1988 | 1,889 | 13.6 | 9,445 | 13.6 | 5,966 | 13.1 | 29,655 | 13.0 |
| 1989 | 1,844 | 13.2 | 9,220 | 13.2 | 5,836 | 12.8 | 29,034 | 12.8 |
| 1990 | 1,925 | 13.8 | 9,625 | 13.8 | 6,181 | 13.5 | 30,763 | 13.5 |
| 1991 | 1,708 | 12.3 | 8,540 | 12.3 | 5,995 | 13.1 | 29,835 | 13.1 |
| 1992 | 1,563 | 11.2 | 7,815 | 11.2 | 5,452 | 11.9 | 27,157 | 11.9 |
| 1993 | 1,299 | 9.3 | 6,495 | 9.3 | 4,311 | 9.4 | 21,490 | 9.5 |
| **Type of municipality** | | | | | | | | |
| Capital area | 3,451 | 24.8 | 17,255 | 24.8 | 20,224 | 44.3 | 100,616 | 44.3 |
| Large urban area | 3,308 | 23.7 | 16,540 | 23.7 | 18,682 | 40.9 | 92,935 | 40.9 |
| Towns or rural area | 7,171 | 51.5 | 35,855 | 51.5 | 6,781 | 14.8 | 33,736 | 14.8 |
| **Country of origin** | | | | | | | | |
| Afghanistan | 2,008 | 14.4 | | | 2,425 | 5.3 | | |
| Iran | 639 | 4.6 | | | 2,260 | 4.9 | | |
| Iraq | 3,165 | 22.7 | | | 12,593 | 27.6 | | |
| Somalia | 1,691 | 12.1 | | | 3,401 | 7.4 | | |
| Yugoslavia | 3,478 | 24.9 | | | 15,139 | 33.1 | | |
| Other countries | 2,982 | 21.4 | | | 9,869 | 21.6 | | |
| **Age at immigration** | | | | | | | | |
| 0–5 years old | 4,160 | 29.8 | | | 16,256 | 35.6 | | |
| 6–10 years old | 5,607 | 40.2 | | | 13,134 | 28.7 | | |
| 11–15 years old | 3,462 | 24.8 | | | 10,078 | 22.1 | | |
| 16–17 years old | 734 | 5.3 | | | 6,219 | 13.6 | | |
| **Accompanying family** | | | | | | | | |
| Unaccompanied | 531 | 3.8 | | | 2,995 | 6.6 | | |
| One parent | 2,548 | 18.2 | | | 7,682 | 16.8 | | |
| Both parents | 10,884 | 77.9 | | | 35,010 | 76.6 | | |

*Notes*: Refugees obtained residence in Denmark/Sweden before 2010 (aged 0–17) with refugee status or through family reunification with a refugee. Majority population were born in Denmark/Sweden to a parent also born in Denmark/Sweden and was matched 1:5 on sex, birth cohort and municipality type. Common mental disorders were measured as psychiatric hospital contacts (inpatient or specialised outpatient) with a main diagnosis related to ICD-10 codes F32-33 or F40-43 or the purchase of a prescribed anti-depressant (ATC code N06A) during 2009–2011.

and the majority population across birth years, while there was substantial heterogeneity by birth year in terms of how LMM developed over the four years in Denmark.

While the trends were largely the same when classifying students with enrolment data in Denmark (see S1 Fig), levels of LMM among refugees were a bit higher.

**Table 2. Completion of upper secondary education* by refugee youth and their majority peers in Denmark and Sweden from 2011–2015.**

| Year | DENMARK | | | | SWEDEN | | | |
|---|---|---|---|---|---|---|---|---|
| | Refugees | | Majority | | Refugees | | Majority | |
| | No. | % | No. | % | No. | % | No. | % |
| 2011 | 4559 | 32.7 | 33871 | 48.6 | 29053 | 63.6 | 181687 | 79.9 |
| 2012 | 5738 | 41.2 | 40851 | 58.7 | 32408 | 70.9 | 202297 | 89.0 |
| 2013 | 6797 | 48.8 | 47607 | 68.4 | 34783 | 76.1 | 206043 | 90.7 |
| 2014 | 7620 | 54.7 | 52107 | 74.8 | 36031 | 78.9 | 207491 | 91.3 |
| 2015 | 8215 | 59.0 | 54510 | 78.3 | 36955 | 80.9 | 208526 | 91.7 |

* Upper secondary school is assumed completed if highest achieved educational level corresponds to ISCED level 3 or above. Highest achieved educational status is measured on 1 October in Denmark and 31 December in Sweden during the corresponding years.

## Mediation analysis

Table 3 presents a summary of the results of the mediation analyses. The total effect corresponds to the overall relative inequality in labour market marginalisation between refugees and majority youth across ages ($OR_{DK}$ = 2.09; 95% CI = [2.02–2.17] and $OR_{SE}$ = 2.31; 95% CI = [2.27–2.36]). The mediation analysis showed that if refugee and majority youth were

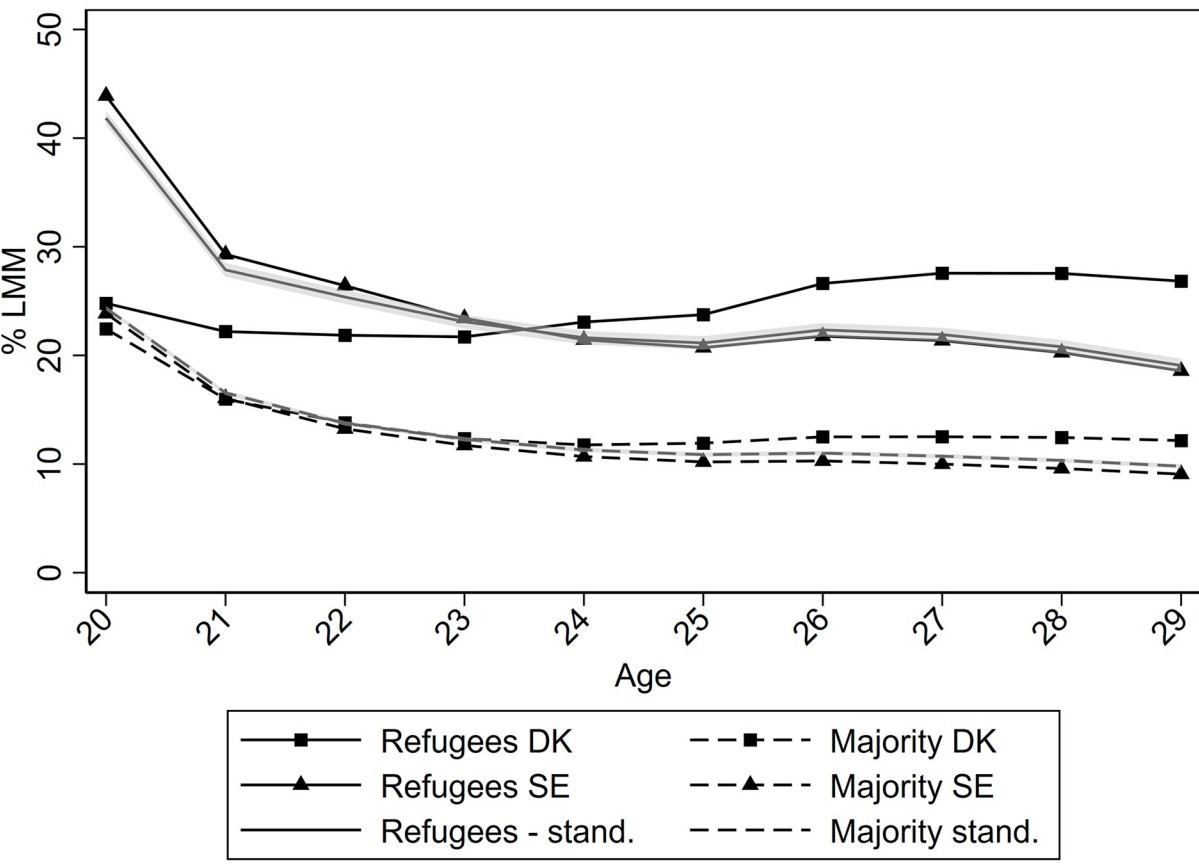

**Fig 1. The proportion experiencing labour market marginalisation (LMM) from age 20–29 in refugees and the majority population in Denmark and Sweden; standardisation using the population distribution in Denmark and parameter estimates from Sweden, with 95% confidence intervals.**

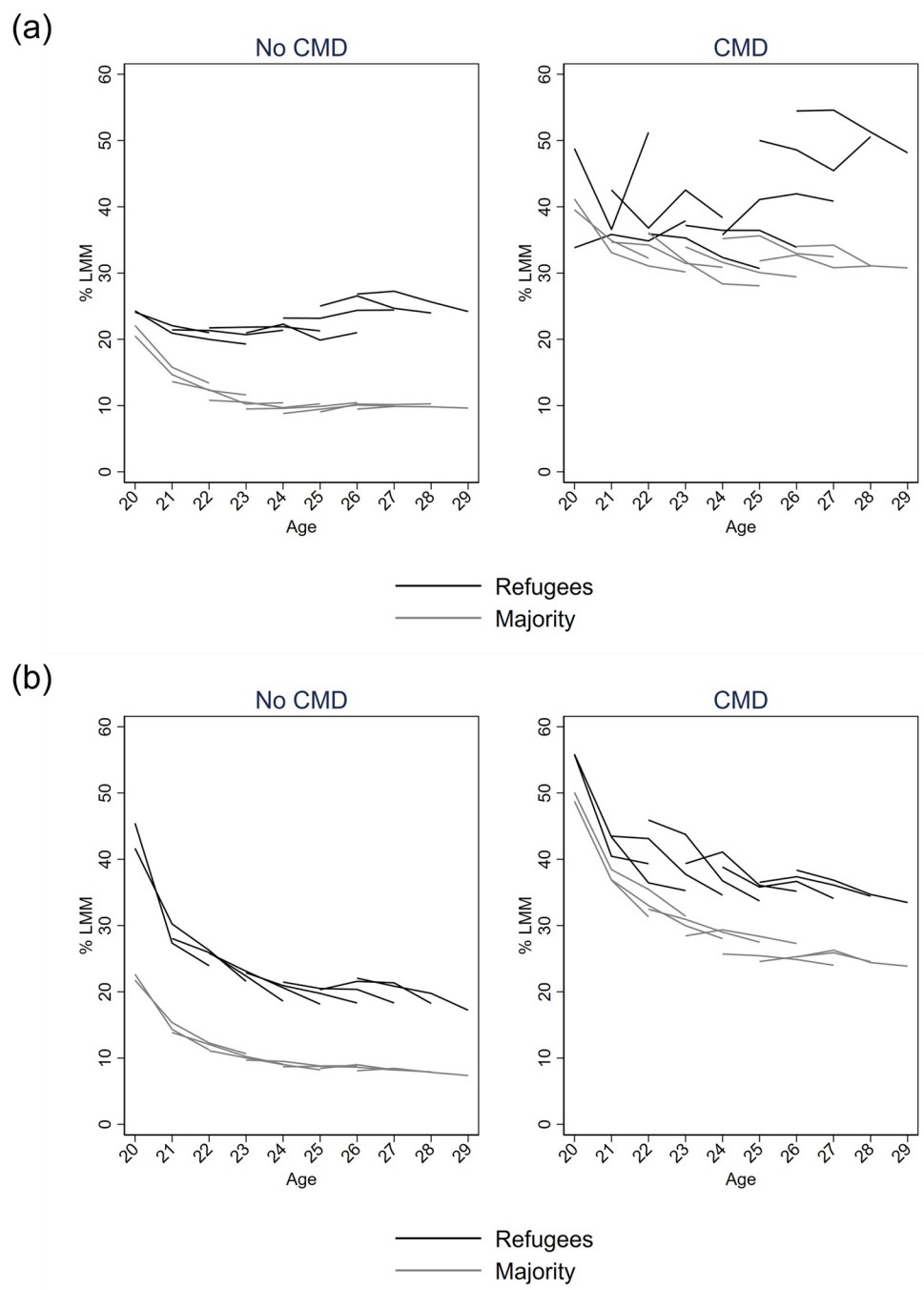

**Fig 2. Proportion of youth from each birth year experiencing labour market marginalisation (LMM) from age 20–29 in Denmark (a) and Sweden (b).** With or without contacts for common mental disorders during 2009–2011. Lines represent repeated observations across birth years, 1986–1993. (A) Denmark. (B) Sweden.

similarly likely to complete secondary school, 77% of the inequality in Denmark and 85% in Sweden would be eliminated. The controlled direct effects after eliminating secondary school completions were 1.25 in Denmark and 1.20 in Sweden. In contrast, there was no mediation by psychiatric contacts for common mental disorders. Including both potential mediators yielded results similar to the analysis with secondary school completions alone.

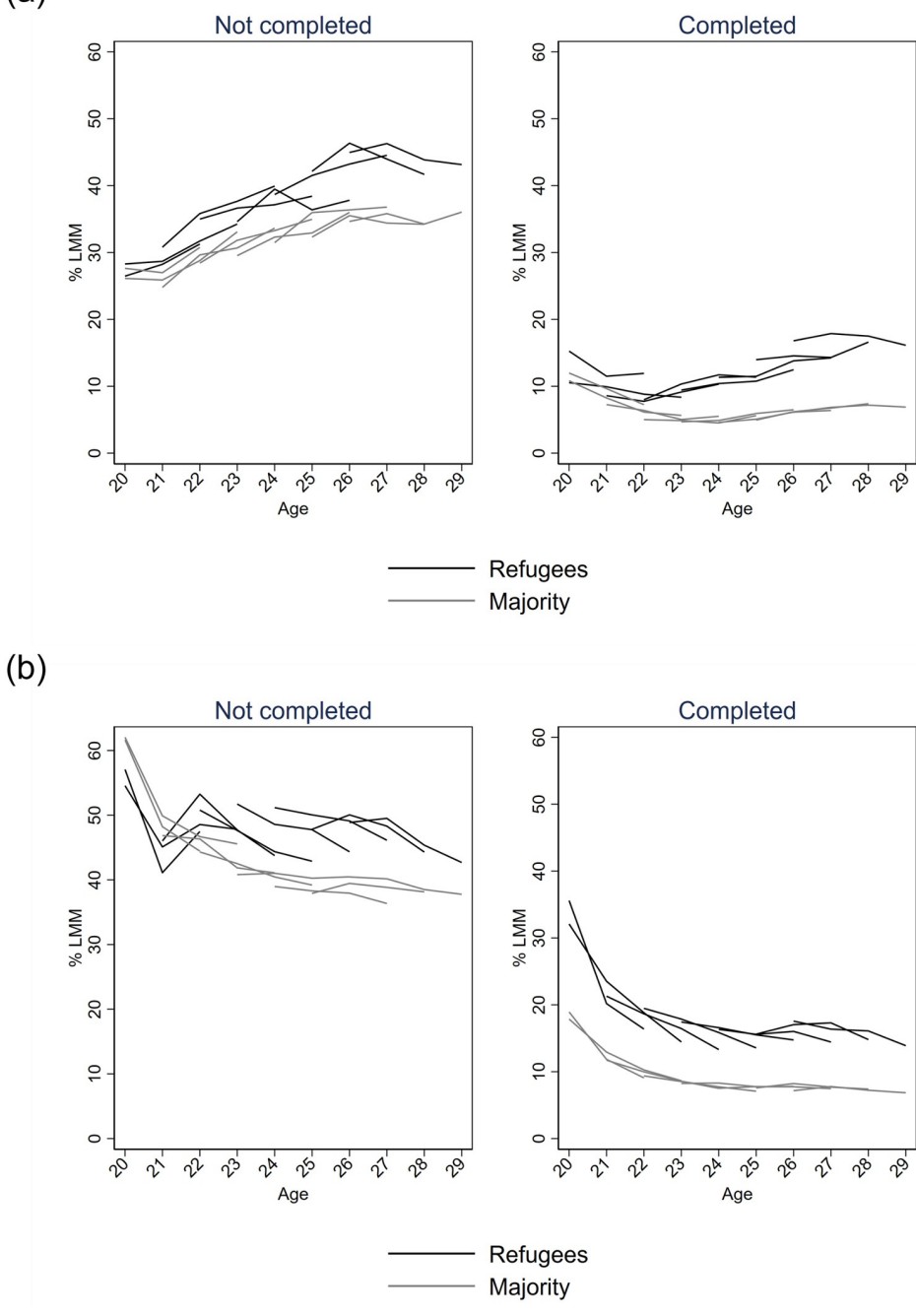

**Fig 3. Proportion of youth from each birth year experiencing labour market marginalisation (LMM) from age 20–29 in Denmark (a) and Sweden (b).** With or without completed upper secondary education each year. Lines represent repeated observations across birth years, 1986–1993. (A) Denmark. (B) Sweden.

Analyses stratified by sex are presented in the S1 Table. The overall trends were the same among boys and girls. However, in Denmark, the overall inequality was largest among boys while the mediating role of secondary school completions was strongest among girls. In Sweden, we found the reverse.

**Table 3. The mediating role of secondary school completion and common mental disorders in the inequality in labour market marginalisation between young refugees and their majority peers.**

| | DENMARK | | SWEDEN | |
|---|---|---|---|---|
| | OR | 95% CI | OR | 95% CI |
| **Model 1: Mediation by secondary school completion** | | | | |
| Total effect | 2.09 | [2.02–2.17] | 2.31 | [2.27–2.36] |
| Controlled direct effect | 1.25 | [1.19–1.32] | 1.20 | [1.16–1.25] |
| Portion eliminated | 1.67 | [1.61–1.73] | 1.92 | [1.89–1.96] |
| Proportion eliminated (%) | 77% | | 85% | |
| **Model 2: Mediation by common mental disorders** | | | | |
| Total effect | 2.09 | [2.02–2.17] | 2.31 | [2.27–2.36] |
| Controlled direct effect | 2.20 | [1.19–1.32] | 2.46 | [2.40–2.51] |
| Portion eliminated | 0.95 | [0.91–0.98] | 0.94 | [0.92–0.96] |
| Proportion eliminated (%) | - | | - | |
| **Model 3: Mediation by both** | | | | |
| Total effect | 2.09 | [2.02–2.17] | 2.31 | [2.27–2.36] |
| Controlled direct effect | 1.38 | [1.30–1.46] | 1.32 | [1.26–1.37] |
| Portion eliminated | 1.52 | [1.46–1.57] | 1.75 | [1.72–1.79] |
| Proportion eliminated (%) | 65% | | 76% | |

*Notes*: Labour market marginalisation: labour market income of less than 12.5% of the median labour market income in the population aged 20–64 and not a student and is measured 2012–2016. Secondary school corresponds to a completed education at ISCED level 3 or above during the year prior to measurement of marginalisation (2011–2015) on 1 October in Denmark and 31 December in Sweden. Common mental disorders are psychiatric hospital contacts (inpatient or specialised outpatient) with a main diagnosis related to ICD-10 codes F32-33 or F40-43 or the purchase of a prescribed anti-depressant (ATC code N06A). Sex, birth year and municipality type were entered as covariates in the relationship between population group and mediator. Proportion eliminated omitted if the portion eliminated differs in direction from the controlled direct effect. Total effect corresponds to the overall inequality between the two groups.

## Discussion

There are two main explanation for the notable country differences in LMM patterns among young refugees and their majority peers in Denmark and Sweden. The first is that the refugee populations differ in terms of important predictors for marginalisation such as socio-economic status of families before flight, health at arrival, and adversities faced prior to and during flight. The second is that host-country factors structure life course patterns among young refugees differently in Denmark and Sweden. The analysis addressed the former mechanism by standardising results for observed characteristics. While these may not account for all relevant differences in the refugee populations, if the findings were explained by composition alone we would expect the observed differences to matter more than they appeared to do.

The findings therefore raise a number of questions concerning the circumstances faced by refugee youth in Denmark and Sweden. The refugee youth observed in Denmark have grown up with swiftly changing and increasingly restrictive immigration and integration policies, as described in the introduction, in increasing contrast to policies in Sweden. Meanwhile, in terms of labour market marginalisation, the younger cohorts of refugees in Denmark appear to be in a better position than older cohorts of refugees. This could suggest that institutional changes other than restrictive policies may be defining for refugee children's life chances. Sweden has a longer history as an immigration destination and has for decades provided asylum for proportionally greater numbers of refugees, but both societies have become increasingly

diverse over the past decades [29]. One perspective that merits further exploration is therefore how capacity to create favourable conditions for refugee children relate to the scale and history of refugee immigration. The volatility between birth cohorts and over time in Denmark, in contrast with the clear and stable trends in Sweden that mirror the majority population (although with a level difference), could indicate that as experience accrues, perhaps most crucially at the school and municipal levels, models of support are developed and refined and the skills of frontline workers are honed.

The strong mediating role of secondary school completion in both countries underscores the critical importance of this phase of schooling. Several features of the educational system in Sweden that have been highlighted to make a difference for young refugees' educational careers in international comparisons, such as the absence of ability-tracking and the availability of alternative pathways to higher education for refugee youth [18, 36], do not stand out in the comparison with Denmark. In both countries, the considerable discretion afforded local service providers means that there is substantial regional variation in how refugee children and families are supported in schools, as well as with housing, language support, job training, and activities involving civil society organizations [37]. In Denmark, however, and unlike Sweden, young refugees are not by default incorporated into the mainstream educational system during the asylum phase, which could present a significant lost opportunity to minimise the amount of time young refugees spend outside stable education. In addition, any student who has (partially) attended a reception class in Sweden is entitled to a support person as they transition into the regular system [36], and within two months schools are obliged by law to develop study plans for each refugee pupil arriving during the final years of compulsory school detailing how the school will ensure that they are ready to enter upper secondary school. Further research should clarify the role of such support arrangements within schools and in the transition between educational levels.

The fact that the labour market inequalities between refugees and their peers did not appear to be mediated by our mental health indicator, psychiatric contacts and prescribed medicine related to CMD, may appear surprising. It is well-established that refugee children and youth are vulnerable to CMD and that they often face challenges within the educational system as a consequence of both this vulnerability and other disadvantages accompanying forced displacement [38]. However, studies of healthcare usage in both Denmark and Sweden have raised concerns that refugee children and youth may not be receiving adequate support in the psychiatric healthcare systems [39, 40]. The lower rates of CMD contacts among refugees in our datasets are consistent with such concerns. While this measure did indicate vulnerability, it did not capture a vulnerability that differentiated refugees from their peers.

The choice of disorders to include in the CMD category may also be important. Population studies have highlighted especially anxiety disorders and post-traumatic stress disorders as high prevalence disorders among young refugees [41, 42]. These were included as CMDs, but the distribution between disorders was not the same among refugee and majority youth. Among youth with a CMD contact, contacts related to stress and in particular post-traumatic stress disorder were more frequent among refugees than their majority peers, with similar rates in Denmark and Sweden. A greater proportion of youth in the CMD group in Sweden had a contact related to depressive or anxiety disorders than in Denmark, but in both countries the level among refugees was slightly below the level among majority youth. In all groups, around 90% of youth with registered CMD had purchased antidepressant medication, except for refugees in Sweden for whom it was around 70%. This suggests that there may be systematic differences in the kind of morbidity that is identified by CMD contacts both between refugees and their majority peers, and between Denmark and Sweden. Trends may also be different for other kinds of disorders, e.g. psychotic disorders, which are less common and may entail different healthcare pathways than the disorders included in this study.

## Strengths and limitations

The analysis builds on high quality registry data from Denmark and Sweden. This has made it possible to develop comparable measures and to identify all refugees in the age-range in question, using juridical information on grounds for residence. Registry-based information also has the advantage that it does not suffer from attrition typical of surveys, an important strength in a study with repeated measures over several years.

Building on a secondary data source, the analysis was limited by the availability of information in the registers. Data on actual morbidity was not available, but was proxied by healthcare use. Enrolment data was not available in Sweden and the high rates of LMM observed especially in Sweden for the youngest members of the sample must be interpreted in light of how stipends are awarded for students below tertiary education. Likewise, important aspects of the composition of the refugee population, such as the educational level and labour market position of parents in the origin countries, were not available. Information on unregistered migration was also not available, which may play a differential role across population groups and countries.

## Conclusions

Young refugees who arrived in Denmark and Sweden as children were more likely to face labour market marginalisation than their majority peers. While the risk was elevated among youth with registered common mental disorders in all populations, such contacts did not capture the mental health vulnerability that differentiated refugees from their majority peers. Studies of determinants of inequalities in the life courses of refugee youth and their peers should therefore be cautious about contact-based measures of CMD.

The risk of LMM decreased with age across birth cohorts in Sweden, while substantial differences between birth cohorts were found in Denmark, with no clear trend in the development with age. It deserves notice that while the contrast between Denmark and Sweden in terms of immigration and integration policies increased over calendar time, the differences in the risk of labour market marginalisation among refugee youth diminished over birth cohorts. This may indicate the importance of factors other than restrictive policies at the national level for refugee children's life chances.

In both countries, differences in secondary school completion mediated most of the inequality. Efforts to narrow the inequality should give special attention to this phase of schooling and to the transition between compulsory and upper secondary school.

## Supporting information

**S1 Fig. The proportion experiencing labour market marginalisation (LMM) from age 20–29 in refugees and the majority population in Denmark.** Students identified using stipend data or enrolment data.
(PDF)

**S1 Table. The mediating role of secondary school completion and common mental disorders in the inequality in labour market marginalisation between young refugees and their majority peers.** Stratified by sex.
(PDF)

## Acknowledgments

We thank Pontus Josefsson for his help with data management.

## Author Contributions

**Conceptualization:** Christopher Jamil de Montgomery, Marie Norredam, Allan Krasnik, Jørgen Holm Petersen, Emma Björkenstam, Lisa Berg, Anders Hjern, Marit Sijbrandij, Peter Klimek, Ellenor Mittendorfer-Rutz.

**Data curation:** Christopher Jamil de Montgomery, Emma Björkenstam.

**Formal analysis:** Christopher Jamil de Montgomery.

**Funding acquisition:** Ellenor Mittendorfer-Rutz.

**Methodology:** Christopher Jamil de Montgomery, Jørgen Holm Petersen.

**Supervision:** Ellenor Mittendorfer-Rutz.

**Validation:** Christopher Jamil de Montgomery, Ellenor Mittendorfer-Rutz.

**Writing – original draft:** Christopher Jamil de Montgomery.

**Writing – review & editing:** Christopher Jamil de Montgomery, Marie Norredam, Allan Krasnik, Jørgen Holm Petersen, Emma Björkenstam, Lisa Berg, Anders Hjern, Marit Sijbrandij, Peter Klimek, Ellenor Mittendorfer-Rutz.

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
