## [Decision Letter · Decision Letter 0]

30 Dec 2021

PONE-D-21-19639Labour market marginalisation in young refugees and their majority peers in Denmark and Sweden: The role of common mental disorders and secondary school completionPLOS ONE

Dear Dr. Jamil de Montgomery 

Thank you for submitting your manuscript to PLOS ONE. After careful consideration, we feel that it has merit but does not fully meet PLOS ONE’s publication criteria as it currently stands. Therefore, we invite you to submit a revised version of the manuscript that addresses the points raised during the review process. Plaease address the points raised by the reviewer particularly the rigour of detrmining the influence of mental health issues in the analysis. 

Please submit your revised manuscript by Feb 13, 2022  If you will need more time than this to complete your revisions, please reply to this message or contact the journal office at plosone@plos.org. Please include the following items when submitting your revised manuscript:A rebuttal letter that responds to each point raised by the academic editor and reviewer(s). You should upload this letter as a separate file labeled 'Response to Reviewers'.A marked-up copy of your manuscript that highlights changes made to the original version. You should upload this as a separate file labeled 'Revised Manuscript with Track Changes'.An unmarked version of your revised paper without tracked changes. You should upload this as a separate file labeled 'Manuscript'.

We look forward to receiving your revised manuscript.

Kind regards,

Gerard Hutchinson, MD

Academic Editor

PLOS ONE

Journal Requirements:

2. In ethics statement in the manuscript and in the online submission form, please provide additional information about the records used in your retrospective study. Specifically, please ensure that you have discussed whether all data were fully anonymized before you accessed them and/or whether the IRB or ethics committee waived the requirement for informed consent. If participants provided informed written consent to have data from their medical records used in research, please include this information.

Reviewers' comments:

Reviewer's Responses to Questions

**Comments to the Author**

1. Is the manuscript technically sound, and do the data support the conclusions?

Reviewer #1: Yes

2. Has the statistical analysis been performed appropriately and rigorously? 

Reviewer #1: Yes

3. Have the authors made all data underlying the findings in their manuscript fully available?

Reviewer #1: No

4. Is the manuscript presented in an intelligible fashion and written in standard English?

Reviewer #1: Yes

5. Review Comments to the Author

Reviewer #1: Well written paper on an important topic.

Minor suggestions to consider.

1) Rationale for using dichotomous indicators for labour market marginalisation is missing. Authors refer to the paper by Bäckman et al, but it would important to explain to readers why certain cut-offs (e.g., "labour market income of less than 12.5% of the median") were chosen.

2) Although common mental disorders typically refer to anxiety and depressive disorders, did authors consider also analysing substance abuse or psychotic disorders?

3) Was there any systematic difference in results between inpatient or outpatient versus antidepressants? I would assume that antidepressant would capture also less severe cases.

6. PLOS authors have the option to publish the peer review history of their article (what does this mean?). If published, this will include your full peer review and any attached files.

Reviewer #1: **Yes: **Christian Hakulinen

---

## [Author Response · Author response to Decision Letter 0]

7 Jan 2022

Response to reviewers related to manuscript PONE-D-21-19639

Dear Editor and Reviewer,

Thank you for taking the time to process and review our manuscript and for the comments provided. In the following, we respond to each comment.

1. Rationale for using dichotomous indicators for labour market marginalisation is missing. Authors refer to the paper by Bäckman et al, but it would important to explain to readers why certain cut-offs (e.g., "labour market income of less than 12.5% of the median") were chosen.

Response: Thank you for this comment. We agree these methodological choices could be made more transparent to the reader. We have therefore added the following sentences to the methods section (new text in italics): 

“In Bäckman et al.’s definition, 12.5% of the median labour market income is a pragmatic cut-off between labour market positions defined as no or negligible labour market participation and part-time or unstable work. While a full-time income at a low salary corresponds approximately to 87.5% of the median labour market income, 12.5% represents an annual income below what would be earned in 2 months in such a position. Additionally, those with an income from disability pensions of at least 25% of the median labour market income (which corresponds to approximately half a year of full-time disability pension) and those registered as unemployed for at least 180 days during the year (more than half the year) were also included in the LMM category. Although the precise cut-off values to a certain extent are arbitrary, these definitions have been used in several other Nordic comparative studies of labour market marginalization (31,33,34) and aligning the definitions across studies facilitates comparisons.”

Reference 31 was already included on the list of references, while references 33 and 34 have been added:

“33. Lorentzen T, Bäckman O, Ilmakunnas I, Kauppinen T. Pathways to adulthood: Sequences in the school-to-work transition in Finland, Norway and Sweden. Soc Indic Res. 2019;141(3):1285–305. 

34. Jakobsen V, Korpi T, Lorentzen T. Immigration and integration policy and labour market attainment among immigrants to Scandinavia. Eur J Popul. 2019;35(2):305–28.”

2. Although common mental disorders typically refer to anxiety and depressive disorders, did authors consider also analysing substance abuse or psychotic disorders?

Response: It is very probable that the trends would be different for other disorders. As our project was originally conceived (and funded!), our focus was on common mental disorders specifically, and our study is one among several using this definition (including depressive disorders and purchases of anti-depressives, anxiety disorders, obsessive-compulsive disorders, and disorders related to severe stress). Our findings have however led us to extend our focus to other disorders, such as psychotic disorders, for future papers. Care pathways are plausibly different for psychotic disorders due to severity, and we do observe higher rates of contact among young refugees for psychosis-related disorders than among their peers.

We do not find it feasible to include an analysis of psychotic disorders in the present manuscript as it, to us, represents another research question that we want to explore in its own right. But we have added the following sentence to the discussion (page 18): “Trends may also be different for other kinds of disorders, e.g. psychotic disorders, which are less common and may entail different healthcare pathways than the disorders included in this study.”

3. Was there any systematic difference in results between inpatient or outpatient versus antidepressants? I would assume that antidepressant would capture also less severe cases.

Response: This is a good point. We write in the discussion that 90% of youth with a CMD contact had purchased antidepressant medication, except for refugees in Sweden for whom it was 70% (page 17). On the one hand, this means that the results, largely, were driven by youth who had purchased antidepressant medication. On the other hand, it means that youth who had a hospital contact but had not purchased antidepressant medication were a very small sub-group. So even if they did have higher risks of marginalization (as more severe cases) and even if the pattern of utilization was different among refugees and majority youth, their contribution to the overall inequality in labor market marginalization would be small.

We have in fact checked to see whether the distribution of CMD contacts indicated through the patient registry (in and outpatient contacts) and through the pharmaceutical registry (anti-depressants) differed for refugees and majority youth. For all three variables, the original composite variable used in the manuscript, the hospital contact variable and the anti-depressants variable, the proportion with a contact was larger in the majority population than in the refugee population. We have added this information to the methods and results sections:

Methods section (page 7): “As a sensitivity analysis, we also investigated whether contact patterns differed between the populations when considering hospital contacts (in- or outpatient) and antidepressant purchases separately.”

Results section (page 10): “In both Denmark and Sweden, the proportion of youth with CMD was larger in the majority population than in the refugee population. This pattern was consistent whether CMD was defined based on hospital contacts alone, purchases of antidepressants alone, or both combined (not shown). The difference was larger among girls than boys in both countries.”

As we conclude that our registry-based CMD variable does not capture the higher mental health vulnerability of refugees as compared with their peers, it does not seem relevant to add further sub-analyses to the study as neither definition singles out refugees as particularly vulnerable. However, we do find this “non-finding” of the explanatory power of a registry-based CMD variable very relevant to the field as such variables are often used. Indeed, having a CMD-related contact (hospital or medicine) does have explanatory power in terms of labor market marginalization in both the refugee and the majority population; but given differences in utilization of psychiatric services, it does not contribute to explain socio-economic inequality between the two groups of young individuals, even though we know from a host of other studies that refugees are particularly vulnerable to mental distress and disorders.

Other changes

To align with journal requirements, we have also made the following changes to the manuscript:

• Formatting of headings

• Correction of title page formatting

• Key words removed from abstract page

• Funding and competing interests information moved

• Reference style updated

• Figure references updated

• Naming of supplementary files updated and captions added at the end of manuscript

• The ethics statement has been updated to include information about anonymization and informed consent

---

## [Decision Letter · Decision Letter 1]

20 Jan 2022

Labour market marginalisation in young refugees and their majority peers in Denmark and Sweden: The role of common mental disorders and secondary school completion

PONE-D-21-19639R1

Dear Dr. Jamil de Montgomery 

We’re pleased to inform you that your manuscript has been judged scientifically suitable for publication and will be formally accepted for publication once it meets all outstanding technical requirements.

Kind regards,

Gerard Hutchinson, MD

Academic Editor

PLOS ONE

Additional Editor Comments (optional):

Reviewers' comments:

Reviewer's Responses to Questions

**Comments to the Author**

1. If the authors have adequately addressed your comments raised in a previous round of review and you feel that this manuscript is now acceptable for publication, you may indicate that here to bypass the “Comments to the Author” section, enter your conflict of interest statement in the “Confidential to Editor” section, and submit your "Accept" recommendation.

Reviewer #1: All comments have been addressed

2. Is the manuscript technically sound, and do the data support the conclusions?

Reviewer #1: Yes

3. Has the statistical analysis been performed appropriately and rigorously? 

Reviewer #1: Yes

4. Have the authors made all data underlying the findings in their manuscript fully available?

Reviewer #1: No

5. Is the manuscript presented in an intelligible fashion and written in standard English?

Reviewer #1: Yes

6. Review Comments to the Author

Reviewer #1: Thank you for answering all my comments. I have no further comments.

Thank you for answering all my comments. I have no further comments.

7. PLOS authors have the option to publish the peer review history of their article (what does this mean?). If published, this will include your full peer review and any attached files.

Reviewer #1: No

---

## [Editor Report · Acceptance letter]

25 Jan 2022

PONE-D-21-19639R1 

Labour market marginalisation in young refugees and their majority peers in Denmark and Sweden: The role of common mental disorders and secondary school completion 

Dear Dr. de Montgomery:

I'm pleased to inform you that your manuscript has been deemed suitable for publication in PLOS ONE. Congratulations! Your manuscript is now with our production department. 

Kind regards, 

on behalf of

Dr. Gerard Hutchinson 

Academic Editor

PLOS ONE